Effects of platelet-rich plasma on mesenchymal stem cells isolated from rat uterus

Vishnyakova Polina 1 2 vpa2002@mail.ru
Artemova Daria 1 5
Elchaninov Andrey 1 3
Efendieva Zulfiia 4
Apolikhina Inna 1 4
Sukhikh Gennady 1 4
Fatkhudinov Timur 2 5
1 National Medical Research Center for Obstetrics, Gynecology and Perinatology Named after Academician V.I. Kulakov of Ministry of Healthcare of Russian Federation , Moscow , Russia
2 Peoples’ Friendship University of Russia (RUDN University) , Moscow , Russia
3 Pirogov Russian National Research Medical University (RNRMU) , Moscow , Russia
4 I. M. Sechenov First Moscow State Medical University of Ministry of Health of Russia (Sechenov University) , Moscow , Russia
5 Scientific Research Institute of Human Morphology , Moscow , Russia
Sergi Consolato
Electronic publication date: 2020 Nov 30
Publication date: 2020
Volume: 8
Electronic Location ID: e10415
Received 2020 Jun 26; Accepted 2020 Nov 2
Copyright: © 2020 Vishnyakova et al.
Copyright year: 2020
Copyright holder: Vishnyakova et al.
License: This is an open access article distributed under the terms of the Creative Commons Attribution License, which permits unrestricted use, distribution, reproduction and adaptation in any medium and for any purpose provided that it is properly attributed. For attribution, the original author(s), title, publication source (PeerJ) and either DOI or URL of the article must be cited.
License URL: https://creativecommons.org/licenses/by/4.0/

Keywords: Mesenchymal stem cells, Platelet-rich plasma, Rat uterine, Plasma, Endometrium

Funding: President Grant for Government Support of Young Russian Scientists 075-15-2019-1120 Russian Science Foundation 17-15-01419 This study was funded by the President Grant for Government Support of Young Russian Scientists No. 075-15-2019-1120 and the Russian Science Foundation (grant No. 17-15-01419). There was no additional external funding received for this study. The funders had no role in study design, data collection and analysis, decision to publish, or preparation of the manuscript.

==============================
Background

Platelet-rich plasma (PRP), which represents a valuable source of growth factors, is increasingly being applied in regenerative medicine. Recent findings suggest the feasibility of using PRP in the treatment of infertility secondary to refractory thin endometrium. Mesenchymal stem/stromal cells (MSCs) of the endometrium are an essential cellular component responsible for extracellular matrix remodeling, angiogenesis, cell-to-cell communication, and postmenstrual tissue repair. Using a rat model, we examine the effects of autologous PRP on MSCs isolated from the uterus and compare them with the effects of autologous ordinary plasma (OP) and complete growth medium.

Methods

MSCs were isolated from uterine tissues via enzymatic disaggregation. Flow cytometry immunophenotyping of the primary cell cultures was complemented by immunocytochemistry for Ki-67 and vimentin. The ability of MSCs to differentiate in osteo-, chondro-, and adipogenic directions was assessed using differentiation-inducing media. The levels of autophagy and apoptosis markers, as well as the levels of matrix metalloproteinase 9 (MMP9) and estrogen receptor α, were assessed by western blotting.

Results

After 24 h incubation, the proliferation index of the PRP-treated MSC cultures was significantly higher than that of the MSC cultures treated with complete growth medium. PRP treatment elevated production of LC3B protein, an autophagy marker, while OP treatment upregulated the expression of stress-induced protein p53 and extracellular enzyme MMP9. The results indicate practical relevance and validity for PRP use in the treatment of infertility.

Introduction

Platelet-rich plasma (PRP) is a term for collected blood plasma with artificially concentrated platelets (Theoret & Stashak, 2014), along with correspondingly increased amounts of latent growth factors as well as other active substances. Blood platelets contain three types of granules: dense granules, α-granules, and lysosomes (Flaumenhaft & Sharda, 2018); the most abundant are α-granules, which contain a number of active substances including chemokines and growth factors (Yun et al., 2016), for example, platelet-derived growth factors (PDGFs), transforming growth factors (TGFs), insulin-like growth factors (IGFs), vascular endothelial growth factor (VEGF), epidermal growth factor (EGF), and fibroblast growth factors (FGFs) (Lubkowska, Dolegowska & Banfi, 2012). PRP is also rich in fibrin, fibronectin, and vitronectin (Marx, 2019).

Degranulation of platelets upon PRP activation promotes fibrinogen cleavage and formation of the gel-like matrix. The main PRP activators used in laboratory practice are calcium, thrombin, and collagen (Maffulli, 2016; Cavallo et al., 2016). The most common PRP activator is calcium, which acts faster than collagen but slower than thrombin (Kim & Byeon, 2019). PRP activation leads to the immediate release of growth factors that start to act at the site of administration. Platelets release 70% of their total content of growth factors within 10 min of PRP activation with CaCl2, while the remaining 30% are released over the course of 1 h. Moreover, these activated platelets continue to produce extra growth factors. As activated platelets die within approximately 8 h following the stimulus, PRP activation should be carried out immediately before use (Kim & Byeon, 2019).

Platelet-rich plasma was introduced as a potential therapeutic for tissue repair by Marx et al. (1998), who reported enhanced rates of bone formation in the osteoplasty of human mandibular defects upon adding PRP to the milled bone graft. Nowadays, PRP is widely used in cosmetology, dentistry, sports medicine, and surgery (Patel et al., 2016; Yuksel et al., 2014; Maffulli, 2016); it can be injected in soft tissues, mixed with a graft, layered, sprayed, or used as a biological membrane (Civinini et al., 2011).

Platelet-rich plasma exerts a local stimulatory effect on cell growth at the site of administration. The benefits of PRP treatment are currently finding recognition in reproductive technologies. A recent study involving 24 female participants with refractory thin endometrium (5 mm or thinner) or a history of IVF failure demonstrated that a course of three repeated infusions of PRP into the uterine cavity led to significant improvements, with 60% and 54% of the patients successfully entering pregnancy and giving birth, respectively (Frantz et al., 2020). The effectiveness of PRP in the treatment of thin endometrium has previously been reported by Zadehmodarres et al. (2017); the study involved 10 female participants with thin endometrium (7 mm or thinner) who received PRP infusions. After two infusions, endometrial thickness exceeded 7 mm in all patients, which enabled a frozen-thawed embryo transfer (FET) procedure. As a result, 5 (50%) of the patients successfully entered pregnancy (Zadehmodarres et al., 2017). Coksuer, Akdemir & Barut (2019), evaluated PRP treatment of thin endometrium as an alternative to estradiol therapy in patients with a history of three or more failed IVF cycles. The resulting average endometrial thicknesses of 10 and 8 mm in the PRP and estradiol groups, respectively, enabled FET in all cases. PRP therapy afforded higher rates of clinical pregnancy and live births (14% vs 6%), as well as a lower occurrence of miscarriages (3% vs 6%), when compared to estradiol therapy (Coksuer, Akdemir & Barut, 2019). Additionally, in a study by Kim et al. (2019), intrauterine administration of autologous PRP improved implantation and pregnancy outcomes for patients with refractory thin endometrium-associated infertility, indicating profound functional consistency of the effects of PRP.

The endometrium is a dynamic structure composed of simple columnar epithelium with uterine glands, as well as the underlying stroma (Pertschuk, 1990) containing blood vessels, nerves, collagen, and reticular fibers, along with a variety of stromal cells (Aplin, 2018). Isolation of mesenchymal stem/stromal cells (MSCs) from the endometrium (Chan, Schwab & Gargett, 2004; Chan & Gargett, 2006) allows the elucidation of tissue-specific functions and markers of these cells, thus opening new prospects for their usage. As actively proliferating cells, MSCs play an important role in tissue homeostasis of the endometrium—they are involved in extracellular matrix remodeling, angiogenesis, cell-to-cell communication, post-menstrual tissue repair, etc. (Mutlu, Hufnagel & Taylor, 2015; Arutyunyan et al., 2016). Endometrial stromal cells are highly susceptible to the action of PRP (Matsumoto et al., 2005); however, only a few works are devoted to studying the exact mechanisms of PRP action on these cells (Aghajanova et al., 2018). In the perspective of infertility treatment, the use of rat endometrial MSCs forms a good model for studying the action of PRP due to ethical accessibility and sufficient size of the biomaterial, allowing for more cells to be obtained when compared to human pipe endometrial biopsy (Jang et al., 2017).

Experimental animal models are accessible and provide better uniformity of samples (Festing & Altman, 2002). The use of human biomaterial is invariably associated with heterogeneity of anamneses and preliminary treatment regimens, as well as the use of different collection protocols. Moreover, in the case of thin endometrium, diagnostic curettage is strongly contraindicated, being substituted with pipe biopsies that provide very limited sample volumes that are insufficient for comprehensive examination (Maclean et al., 2020).

The uterine tissues were preferred as a source for MSCs over the conventional sources (including red bone marrow, adipose tissue, and the umbilical cord) due to the significant differences in functional properties between MSCs isolated from various locations in the body. For example, MSCs isolated from adipose tissue show higher proliferation rates and a higher adipogenic differentiation capacity compared to MSCs isolated from red bone marrow (Brown et al., 2019). At the same time, bone marrow-derived MSCs are more prone to osteogenic differentiation, whereas MSCs isolated from muscle tissue express myoblastic markers and show the highest rates of differentiation into myogenic progeny (Brown et al., 2019). Despite the common immunophenotype signatures of MSCs isolated from different tissue sources (positive for CD73, CD90, and CD105 expression and negative for the hematopoietic markers CD34 and CD45 on their surface), these cells also exhibit a number of tissue-specific surface markers (Klimczak & Kozlowska, 2016) indicating tissue-dependent functional specificity, for example, SUSD2 and CD146 with co-expression of PDGF-Rβ have been shown to be specific for endometrium (Tempest, Maclean & Hapangama, 2018).

The mechanisms of physiological and reparative regeneration of the endometrium are poorly understood. Therefore, we were interested in studying the cellular mechanisms of endometrial regeneration, as well as its stimulation through endometrial MSC activation via PRP. In this study, we examine the effects of autologous PRP on MSCs isolated from rat uterus and compare them with the effects of autologous ordinary plasma (OP).

Materials and Methods

Ethical disclosure

The authors state that they have obtained appropriate institutional review board approval or have followed the principles outlined in the Declaration of Helsinki for all human or animal experimental investigations. The study was approved by the Ethical Review Board at the Scientific Research Institute of Human Morphology (Protocol No.15, 9th of December, 2019).

Animals

Outbred, 8-week-old female Sprague–Dawley rats (250–300 g) were obtained from the Institute for Bioorganic Chemistry branch animal facilities (Pushchino, Moscow region, Russia). All experimental work involving animals was carried out according to the Standards of Laboratory Practice (National Guidelines No. 267 by Ministry of Healthcare of the Russian Federation, June 1, 2003), and all efforts were made to minimize suffering. The animals were adapted to laboratory conditions (23 °C, 12 h/12 h light/dark, 50% humidity, ad libitum access to food and water) for 2 weeks prior to experimentation. In adult female rats, estrous cycle stage was determined by taking a vaginal smear. Vaginal smears were stained with methylene blue and stage was determined via assessment of the cellular composition. Following euthanasia in a CO2-chamber at the stage of metestrus, the uterus was dissected and blood was collected via puncturing the heart. The animals served only as a source of the uteruses and autologous plasma; therefore, no experimental conditions and endpoints were applied.

PRP and OP preparation

Blood was collected in tubes containing 2 ml of heparin (5,000 IU/ml) and 800 μl of 10% sodium citrate. An average of 6 ml of blood was obtained per animal. PRP was obtained based on a protocol developed by Yazigi Junior et al. (2015). The blood was centrifuged at 400g for 10 min, the plasma was collected in a new tube and centrifuged again at 400g for 10 min at 4 °C. After centrifugation, 70% of the supernatant (the platelet-poor plasma, PPP) was discarded. The remaining fraction, the PRP, which belongs to the L type (L-PRP) according to modern classification, was preserved. The platelet counts were determined on a TC20 Automated Cell Counter (Bio-Rad, Hercules, CA, USA) and constituted 50 × 106 platelets/ml on average. To obtain OP, the blood was centrifuged at 4 °С, 2,000g for 15 min, and the supernatant was collected. The prepared OP and PRP were aliquoted, frozen, and stored at −20 °C. Before use, PRP was activated by adding 10% CaCl2 (10 μl per 200 μl of PRP) according to the protocol by Messora et al. (2011).

Protocol for obtaining MSCs from rat endometrium

The primary cell cultures were obtained based on the protocol by De Clercq, Hennes & Vriens (2017). with modifications. The dissected uterus was minced with scissors in Hank’s Balanced Salt Solution (HBBS) and transferred into a tube containing 0.25% trypsin. The tube was incubated at 4 °C for 1 h, followed by 1 h at 22 °C, and, finally, 15 min at 37 °C under periodic shaking. The supernatant was taken and the solid bulk was transferred to a solution containing collagenases type I and type IV in 0.05% trypsin-EDTA solution (collagenase I:collagenase IV:trypsin-EDTA, 1:1:10), incubated at 37 °C for 30 min, and passed through a 70 μm cell strainer. The material remaining on the strainer was transferred to a fresh solution of collagenases with 0.05% trypsin-EDTA, incubated at 37 °C for 15 min, and passed through a strainer again. The resulting suspension, which contained the cells of interest, was centrifuged at 300g for 5 min at 22 °C. Then, the pellet was resuspended in HBBS supplemented with 1% FBS, centrifuged an additional time, and resuspended in complete growth medium (CGM; DMEM/F-12 supplemented with 10% FBS, L-glutamine, and penicillin/streptomycin) for cultivation. The obtained cells were verified for compliance with the minimal criteria for MSCs issued by the International Society for Cellular Therapy (Dominici et al., 2006) (adhesion to untreated plastic, specific profile of surface antigens, and in vitro differentiation towards osteogenic, chondrogenic, and adipogenic progeny). The effects of PRP and OP on MSCs were assessed by culturing the cells for 24 h in medium supplemented with either 10% PRP or 10% OP in place of FBS. CGM with 10% FBS was used as the control.

Flow cytometry analysis

Immunophenotyping of the cells for surface and intracellular markers was performed upon reaching 80% confluence of the cultures. The harvested cells were centrifuged at 800g for 10 min, the supernatant was discarded, and then the cells were fixed in 2% paraformaldehyde for 15 min at room temperature (RT), diluted with 5 ml of PBS, and centrifuged at 1,500g for 10 min. The pellet was then resuspended in 1 ml of PBS. For immunostaining, 1 × 105 fixed cells were incubated in 100 μl of Rinsing Solution (Miltenyi Biotec, Somerville, MA, USA) with primary antibodies to CD90 (ab225, 1/100, Abcam), CD45 (130-107-846, clone REA504, 1/20, Miltenyi Biotec, Somerville, MA, USA), CD105 (ab107595, 1/100, Abcam), and CD34 (PAB18289, 1/100, Abnova) at RT for 1 h. Subsequently, the cells were incubated with secondary antibodies: anti-mouse Ig-FITC (ab6785, 1/500, Abcam) or anti-rabbit Ig-PE (sc-3739, 1/100, Santa Cruz) at RT for 1 h in the dark. After the incubation, the cells were washed in PBS, resuspended in 0.5 ml PBS, and then transferred to fresh tubes for analysis using a FACScan flow cytometer (Becton Dickinson, Franklin Lakes, NJ, USA) with CellQuest software.

Induced differentiation of MSCs

The ability of MSCs to differentiate in osteo-, chondro-, and adipogenic directions was assessed at early passages (up to 7). The cells were grown to 70% confluency in CGM, and then the growth medium was replaced with differentiation medium (for differentiation) or CGM (for the controls). Differentiation into adipogenic progeny was accomplished using the StemPro® Adipogenesis Differentiation Kit (Gibco, Amarillo, TX, USA) over the course of 7 days. At the end of the differentiation process, the cells were fixed with ethyl alcohol:formalin solution (1:4) for 3 min and then stained with Sudan III (5.7 mM) for 10 min, in order to visualize fat droplets. For osteogenic differentiation, the medium was supplemented with dexamethasone (10−7 M) and ascorbic acid (0.2 mM) over the course of 2 weeks. At the end of the supplementation period, the cells were fixed with 70% ethanol. For the detection of mineralization sites, the cells were stained with 40 mM alizarin red S solution (pH = 4.7) for 5 min. Chondrogenic differentiation was performed using the StemPro® Chondrogenesis Differentiation Kit (Gibco, Amarillo, TX, USA). Following 2 weeks of differentiation, the cells were fixed with 4% formalin for 1 h. For detection of mucopolysaccharides, the cells were stained with 1% alcian blue for 24 h. Samples were analyzed using an Axiovert 40 CFL inverted microscope (Zeiss, Oberkochen, Germany) using ZEN software (Carl Zeiss, Oberkochen, Germany).

Immunocytochemistry

The MSCs were grown on glass coverslips (Fisher Scientific, Waltham, MA, USA) coated with gelatin and placed in Petri dishes (35 × 10 mm). For immunocytochemistry, the cells were fixed in 2% paraformaldehyde. The coverslips with fixed cells were treated with 0.1% Triton X-100 for 10 min (cell membrane permeabilization) and washed with PBS. Non-specific binding sites were blocked with 1% BSA in PBS with 0.1% Tween-20 for 30 min. The coverslips were incubated with either antibodies to Ki-67 (ab15580, 1/100, Abcam) or vimentin (ab8978, 1/250, Abcam) at 4 °C for 24 h. After incubation, the coverslips were washed with PBS and then incubated with secondary anti-rabbit-PE antibodies (1/200) or anti-rabbit-FITC antibodies (1/200) in the dark at RT for 1 h and, subsequently, washed with PBS. To stain the nuclei, the coverslips were incubated with DAPI (0.004 mg/ml) at 37 °C for 10 min. Next, the coverslips were washed in PBS and mounted in Aqua-Poly Mount (Polysciences, Warrington, PA, USA). Photographs were taken using a Leica DM 4000B fluorescence microscope (Leica Microsystems, Wetzlar, Germany) with LAS AF v.3.1.0 software (Leica Microsystems, Wetzlar, Germany).

Western blot assay

The cells were washed with PBS and lysed in ice-cold RIPA buffer. The sample was mixed with 2X Loading Buffer and incubated at 95 °C for 1 min. The samples were stored at −80 °C until use and heated for 1 min at 95 °C before loading. The proteins were separated by 10–12.5% sodium dodecyl sulfate polyacrylamide gel electrophoresis (SDS-PAGE) and transferred from the gel to PVDF membranes via the semi-wet approach using the Trans-Blot® Turbo™ RTA Mini LF PVDF TransferKit (Bio-Rad Laboratories, Inc., Hercules, CA, USA). The membranes were blocked with 5% milk in Tris-buffered saline containing 0.1% Tween (TTBS) at RT for 1 h, then stained overnight with primary antibodies to LC3B (1:3,000, ab48394, Abcam), Bcl-2 (1:1,000, ab32124, Abcam), p53 (1:250, ab90363, Abcam), ERα (1:2,000, ab3575, Abcam), MMP-9 (1:1,000, ab38898, Abcam), and GAPDH (1:500, sc-25778, Santa Cruz). Subsequently, the cells were stained with HRP-conjugated secondary antibodies (1:5,000, Bio-Rad Laboratories, Inc., Hercules, CA, USA). The chemiluminescent signal was developed using the Novex ECL Kit (Invitrogen, Waltham, MA, USA) and visualized on a C-DiGit® Blot Scanner (LI-COR, Lincoln, NE, USA) using Image Studio™ Acquisition Software (LI-COR, Lincoln, NE, USA). A figure of an uncropped membrane is available in the Supplemental Information (Fig. S1). Relative protein levels were determined via normalization to GAPDH signals. The bands represent biological replicates (i.e. correspond to different individuals).

Statistical analysis

Statistical data processing was performed using GraphPad Prism 8 software (GraphPad Software, San Diego, CA, USA). The Shapiro–Wilk test was applied to assess normality of the distributions. In cases of a normal distribution, one-way ANOVA with Turkey post-hoc test for multiple comparison was used. In cases of a non-normal distribution, the Kruskal–Wallis test with post-hoc Dunn test was used. Differences were considered statistically significant at p < 0.05.

Results

Characterization of the isolated MSCs

The uterus is a non-classical source of MSCs. Given the wide variability of MSC phenotypes and tissue-specific features (Kwon et al., 2016), the first stage of our study was to demonstrate compliance of our rat uterine mesenchymal cell cultures under the established minimal criteria for MSCs, which includes immunophenotyping and demonstrating the capability of induced in vitro differentiation into different mesenchymal lineages. The obtained cultures of fibroblast-like cells were immunophenotyped using flow cytometry for CD90, CD105, CD45, and CD34. The region of interest was selected in relation to forward and side scattering values in an FSC-SSC dot plot diagram (Fig. 1A), which reflects the size and granularity of the cells, respectively. We gated the major pool of single cell events (R1) excluding debris (Fig. 1A). The peaks of fluorescence were distinctly shifted in histograms for the MSC samples stained specifically with antibodies to CD90 and CD105 when compared to controls stained with only secondary antibodies (Fig. 1B, green curve). We found that 72.7% of the cells were positive for CD90 and 31.8% of the cells were positive for CD105, indicating correspondence of the obtained cultures to the CD90+CD105+ phenotype. Interestingly, 3.1% and 20.1% of the cells were positive for CD45 and CD34, respectively. To further confirm the compliance, we stained the cultures for vimentin. The immunocytochemical assay revealed the presence of vimentin protein in 100% of the cells (Fig. 1C).

Figure 1 Characterization of the mesenchymal cell cultures isolated from rat uterus.

Flow cytometry analysis: (A) representative forward and side scattering dot plot with the region of interest (R1). The percentages of CD90, CD105, CD45, and CD34 positive cells in R1 are indicated (B). Green curve corresponds to the control (stained with secondary antibodies only). Anti-vimentin staining (C): negative control (secondary antibodies only) and immunocytochemistry with anti-vimentin antibodies (red) and the nuclei counterstained with DAPI (blue) are presented. Bars, 20 μm. Induced cell differentiation assay (D): control cells and cells following the induction of differentiation are presented. Adipogenic differentiation was revealed by Sudan III staining, osteogenic differentiation was revealed by alizarin red staining, and chondrogenic differentiation was revealed by alcian blue staining. Bars, 50 μm. One experimental block is marked by a dotted line.

We promoted differentiation of the cell cultures into adipogenic, osteogenic, and chondrogenic lineages using specific combinations of inducers. The adipogenic differentiation was identified by the formation of lipid droplets, which were revealed by staining with Sudan III (Fig. 1D). During osteogenic differentiation, the cells formed characteristic conglomerates, with foci of calcification visible following alizarin red staining (Fig. 1D). Conglomerates were also formed by cells during chondrogenic differentiation, where mucopolysaccharide production was revealed via staining with alcian blue (Fig. 1D).

Effects of autologous PRP and OP on cell proliferation and cell death

Effects of autologous PRP and OP on the rat uterine MSC cultures were evaluated via a 24 h incubation of the cells in medium containing 10% PRP or 10% OP instead of FBS. Accordingly, CGM with 10% FBS was used as a control. Proliferation was assessed by immunocytochemical staining for Ki-67 following the incubation (Fig. 2A). The percentage of Ki-67 positive nuclei divided by the total number of nuclei (proliferation index) was 18.1% for CGM, 41.9% for PRP, and 21.7% for OP (Fig. 2B). With regards to PRP treatment, the differences were statistically significant (p = 0.04).

Figure 2 Assessment of cell viability and cell death following the 24 h incubation of rat uterine mesenchymal stem/stromal cells (MSCs) with complete growth medium (CGM), platelet-rich plasma (PRP), or ordinary plasma (OP).

Immunocytochemical staining (A) for Ki-67 (green) after 24 h exposure of the MSC cultures to CGM, PRP, and OP; the nuclei were counterstained with DAPI (blue). Bars, 20 μm. (B) Proliferation index calculated as the number of Ki-67 positive nuclei divided by the total number of nuclei. *p < 0.05 vs CGM. (C) Representative western blot membranes with the proteins isolated from MSCs after 24 h incubation with the studied agents, stained with p53, Bcl-2, LC3B, and GAPDH specific antibodies. Relative protein levels of p53 (D), Bcl-2 (E), and LC3B (F), normalized by GAPDH level. *p < 0.05 vs CGM group. One experimental block is marked by a dotted line.

To evaluate the activation or inhibition of apoptosis, we analyzed the production of the stress-induced protein p53 and the anti-apoptotic protein Bcl-2. p53 protein level was significantly higher in the cells exposed to OP (p = 0.03) when compared with control (Figs. 2C and 2D). After PRP exposure, the p53 production level did not change. Relative level of Bcl-2 production did not differ among the studied groups (Figs. 2C and 2E).

Autophagy was assessed by analyzing the level of production of the autophagy marker LC3B. Western blot analysis revealed elevated production of LC3B protein in the cells exposed to autologous PRP when compared with control (CGM, Figs. 2C and 2F); the observed effect was statistically significant.

Endometrium receptivity: effects of autologous PRP and OP on matrix metalloproteinase 9 (MMP9) and estrogen receptor α (ERα) production

Zinc-metalloproteinase MMP9 participates in extracellular matrix remodeling. We estimated potential invasiveness of MSCs as MMP9 production. Although elevated levels of MMP9 production were observed in both the OP and PRP groups when compared to CGM, the MMP9 upregulation was only statistically significant for the OP treated cultures (p = 0.03, Figs. 3A and 3B). Additionally, we compared ERα levels, which can be partially associated with endometrial receptivity (Figs. 3C and 3D). Despite elevated ERα levels in the PRP and OP treated cultures, these differences were not significant.

Figure 3 Western blot membrane stained with antibodies to matrix metalloproteinase 9 (MMP9) and estrogen receptor α (ERα).

A representative western blot membrane with the proteins isolated from mesenchymal stem/stromal cells after 24 h incubation with complete growth medium (CGM), platelet-rich plasma (PRP), or ordinary plasma (OP), stained with MMP9 (A), ERα (C), and GAPDH specific antibodies. Relative protein levels of MMP9 (B) and ERα (D), normalized by GAPDH level. *p < 0.05 vs CGM group.

Discussion

The administration of PRP for treatment of thin endometrium has been already introduced into clinical practice in a pilot study by Kim et al. (2019), which aimed at increasing endometrial thickness and, accordingly, the probability of implantation. However, the exact mechanism(s) behind the positive effect of PRP on endometrial thickness remains obscure. In this study, we attempt to identify signaling pathways activated in the multipotent stromal cells of the endometrium under the influence of autologous PRP using rat endometrium as a model. Endometrial MSCs play a key role in the stroma: they participate in the regeneration of the functional layer of the endometrium due to the presence of sex hormone receptors and their ability for extracellular matrix remodeling (Mutlu, Hufnagel & Taylor, 2015). These features emphasize the importance of MSCs in determining receptivity of the endometrium during the implantation period. We isolated primary cultures of stromal cells from rat uterus and proved that these cells were essentially MSCs due to their ability to differentiate towards adipogenic, osteogenic, and chondrogenic lineages in vitro. Phenotypic profiles of the isolated primary cultures were compliant with the CD90+CD105+Vimentin+CD45-CD34− profiles established for MSCs. The presence of cells positive for the CD45 and CD34 markers can be explained by a small admixture of hematopoietic cells and high vascularization of the endometrial stroma. Considering the precedent of 81% CD34− for synovial fluid-derived MSCs and 97.5% CD34− for synovial membrane-derived MSCs cultures classified as CD34 negative, we classify the obtained cultures as CD45 and CD34 negative. The percentage of cells positive for CD105 was lower than expected; however, the published evidence indicates substantial variability of this parameter when assessing MSCs isolated from various tissue sources (Ponnaiyan & Jegadeesan, 2014). In addition, it was shown that the population of MSCs isolated from one source is heterogeneous and can be represented by both CD105− and CD105+ cells (Pham, Vu & Van Pham, 2019). It is important to note that CD105 is a component of the TGFβ receptor, whose level fluctuates depending on the estrous cycle in the rat (Caron et al., 2009).

In the experiments involving plasma treatment, the obtained MSCs were incubated for 24 h in control medium (DMEM/F-12 based CGM with 10% FBS) or medium supplemented with autologous 10% PRP or 10% OP instead of FBS. We observed increased proliferation rates of MSCs under the influence of PRP compared with the influence of FBS. This observation indicates mitogenic effects of PRP on stromal cells. We also studied the protein production levels for the established markers of apoptosis and autophagy. The stress-induced protein p53 is a transcription factor that regulates the cell cycle and acts as a tumor suppressor (Labuschagne, Zani & Vousden, 2018). An important function of p53 is to prevent the accumulation of DNA damage. In the case of damage to cellular DNA, p53 promotes cell cycle arrest and triggers the emergency DNA repair systems; in the case of extensive DNA damage, p53 triggers apoptosis (Wang, Simpson & Brown, 2015). We show that OP promotes increased production of p53 protein by stromal cells, revealing a certain degree of cellular stress associated with the OP treatment. At the same time, both PRP and OP have no effect on the levels of production of the anti-apoptotic Bcl-2 protein by MSCs. Bcl-2 is an anti-apoptotic protein localized on the outer mitochondrial membrane, as well as the membranes of the nuclear envelope and the endoplasmic reticulum (Delbridge et al., 2016). Apparently, Bcl-2 activation is not involved in the effects of PRP and OP.

We used LC3B, a protein which is involved in the biogenesis of autophagosomes (Barth, Glick & Macleod, 2010), as an autophagy marker. Upon binding to the membrane, LC3 conjugates with phosphatidylethanolamine lipid (Tanida, Ueno & Kominami, 2008). Following autophagosome formation, LC3-II is released into the cytosol from the outer layer, while LC3-II located in the inner layer is exposed to hydrolases. LC3B-II is conventionally used as a marker of autophagosomal activity. We observed excessive production of LC3B in MSCs after exposure to PRP. This observation suggests that PRP contributes to self-renewal of the endometrial stromal cells by enhancing autophagy. Rapamycin-induced autophagy was shown to enhance the viability of MSCs, while shRNA-mediated knockdown of autophagy-associated genes decreased their viability (Molaei et al., 2015; Jakovljevic et al., 2018). Decreased autophagy levels are currently considered to be one of the mechanisms underlying the aging of MSCs (Fafián-Labora, Miriam & María, 2019).

Endometrial receptivity is the result of a combination of different characteristics of the endometrium that are responsible for its ability to promote implantation. One indicator of endometrial receptivity is the level of ERα, as this receptor mediates the actions of estrogens preparing the endometrium for implantation via increasing its thickness through increasing the cell proliferation rate. In clinical practice, the level of ERα production can be evaluated by staining the endometrial cells with antibodies to ERα (Glasser et al., 2002), which is expressed by several subpopulations of endometrial stromal cells, including MSCs (Zhou et al., 2001). Therefore, we employed ERα as a marker to reflect the conditional receptivity of the endometrium in our model. We observed no significant differences in the level of ERα production upon exposure of the MSCs to PRP or OP, which may indicate that the beneficial effects of these agents are estrogen-independent. However, this finding should be verified on larger samples to exclude individual variations in the estrogen dependence of the plasma treatment effects.

Aside from being sensitive to the action of hormones, the endometrium should be susceptible and supportive to invasion by the trophoblast. Trophoblast cells bind to the endometrium, proliferate, and penetrate deep into the stroma where they come into contact with maternal blood. Progressive remodeling of the extracellular matrix is a hallmark of this process (Bischof, Meisser & Campana, 2002). The expression of matrix metalloproteinases, observed not only by the trophoblast but also by the endometrial stromal cells, provides effective support to the trophoblast invasion (Bischof, Meisser & Campana, 2002). In this study, we used MMP9 production as an indicator of the MSCs capacity for extracellular matrix remodeling, which is an important parameter of morphogenetic plasticity and receptivity of the endometrium. We demonstrated an increase in MMP9 production specifically facilitated by OP treatment when compared to CGM.

In general, the obtained results indicate stimulatory effects by PRP on the endometrium, as indicated by the behavior of the MSCs isolated from it. Other studies have also indicated that PRP enhances cell proliferation and differentiation in the uterus (Etulain et al., 2018; Aghajanova et al., 2018). Intrauterine administration of PRP to rats with damaged endometrium enhanced proliferation, thus promoting tissue repair and also reducing fibrosis (Jang et al., 2017).

In a study by Kim et al. (2020), therapeutic effects from PRP treatment in a mouse model of Asherman’s syndrome were demonstrated. Asherman’s syndrome is characterized by formation of adhesions and fibrotic lesions inside the uterus. Injections of human PRP at the sites of damage reduced the degree of fibrosis and promoted recovery. It was shown that, after PRP therapy, implantation potential substantially increased and 83.3% of PRP-treated mice gave birth to live offspring, compared with 0.0% in the control group.

The receptivity of the endometrium strongly depends on the inflammatory status and favorable antimicrobial conditions. Pronounced anti-inflammatory effects of PRP on the endometrium were demonstrated in horses by Reghini et al. (2016). Uterine infusions of PRP to mares with chronic endometritis significantly reduced the signs of neutrophilic infiltration and the volume of intrauterine fluid accumulation when observed 24 h after treatment (Reghini et al., 2016). The mechanisms of PRP action on the endometrial cells are still disputable and, apparently, involve anti-inflammatory and pro-proliferative signaling.

The effects of OP on the endometrium are notably understudied. In our experiments, OP induced up-regulation of p53 and MMP9 protein expression, so the beneficial effects of OP are less pronounced. The growth factors released from the PRP activated platelets stimulate proliferation of endometrial cells and enhance autophagy, indicating advantages of PRP as a treatment agent for endometrial dysfunction.

Conclusions

The use of modified blood plasma is expanding in clinical applications. This study evaluates the effects of autologous PRP and OP on tissue-specific MSCs from rat endometrium. Exposure to PRP enhances proliferation of the uterine MSCs, with a significant increase in autophagy. Exposure to OP increased production of the stress-induced protein p53 and the extracellular enzyme MMP9. The results indicate the potency of PRP for the treatment of infertility, particularly in the management of thin endometrium. An understanding of the molecular pathways mediating the beneficial effects of PRP will expand the range of PRP applications.

Supplemental Information

Supplemental Information 1 Western blots membranes.

Raw membranes after chemiluniniscence visualization

Click here for additional data file.

Supplemental Information 2 Raw data.

IHC and western blot counting

Click here for additional data file.

Supplemental Information 3 Full-size membrane after blotting of polyacrylamide gel. Schemes of membrane Ponceau S staining.

Click here for additional data file.

Special thanks to Natalia Usman for her help with proofreading of the manuscript.

Additional Information and Declarations

Competing Interests

Author Contributions

Animal Ethics

Data Availability

The authors declare that they have no competing interests.

Polina Vishnyakova conceived and designed the experiments, performed the experiments, analyzed the data, prepared figures and/or tables, authored or reviewed drafts of the paper, and approved the final draft.

Daria Artemova performed the experiments, analyzed the data, prepared figures and/or tables, authored or reviewed drafts of the paper, and approved the final draft.

Andrey Elchaninov conceived and designed the experiments, performed the experiments, analyzed the data, prepared figures and/or tables, authored or reviewed drafts of the paper, and approved the final draft.

Zulfiia Efendieva performed the experiments, authored or reviewed drafts of the paper, and approved the final draft.

Inna Apolikhina conceived and designed the experiments, analyzed the data, authored or reviewed drafts of the paper, and approved the final draft.

Gennady Sukhikh conceived and designed the experiments, analyzed the data, authored or reviewed drafts of the paper, and approved the final draft.

Timur Fatkhudinov conceived and designed the experiments, analyzed the data, authored or reviewed drafts of the paper, and approved the final draft.

The following information was supplied relating to ethical approvals (i.e., approving body and any reference numbers):

The study was approved by the Ethical Review Board at the Scientific Research Institute of Human Morphology (Protocol No. 15, 9th of December, 2019).

The following information was supplied regarding data availability:

The uncropped membranes after western blotting are available in the Supplemental File.

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
