# Peer review of "Effects of platelet-rich plasma on mesenchymal stem cells isolated from rat uterus"

_PeerJ, doi:10.7717/peerj.10415_

## Round 0.1 · original submission · Major Revisions

The topic will be of interest to the readers of the journal. However, the manuscript, especially the introduction, is poorly written and the ideas do not flow. As a reviewer, I am not able to identify the main objective. In the cover letter, it is mentioned that the molecular mechanism of PRP action on autologous mesenchymal stem cells (MSCs) was performed, but there is no autologous treatment in the report.

The review of the literature is not thorough enough to provide an adequate background of the topic. There are many studies on the effects of platelet-rich plasma on the activity of human or animal-derived MSC (like bovine) that the author did not mention, which could be elaborated on in the discussion section. There is a difference in the writing style between the introduction and the discussion, which might suggest the participation of more than one author in the writing process. I also suggest that the manuscript be edited for English grammar.

Please follow thoroughly all criticisms and comments.

I strongly suggest the authors revise the manuscript and replying point-by-point to the major and minor concerns. If you do not comply with this policy, your manuscript will be rejected.

Reviewer 1 ·

Basic reporting

Overall, the research has an important clinical message and should be of great interest to the readers.
The study outcomes could be outlined as follows:
1)The isolated MSC s from rat uterine met the minimum criteria for MSCs, which suggests that using endometrium as an alternative site for MSC is reasonable.
2)The PRP has a desirable effect on the MSCs behaviors which can be seen in the proliferation rate as measured by the percentage of Ki-67.
3)PRP or OP has an insignificant effect on the ER-alpha or BCL-2.
4)OP treated MSCs have higher levels of MMP-9 expression & p53.
5)Both PRP and OP induce autophagy activity of MSCs.

The study objectives were too broad and should be narrowed down to a specific primary objective. Sufficient background information is not provided to explain what has already been done and more effort should be made to justify the purpose of this study and generate interest in the reader.. The primary hypothesis should be discussed in more detail to provide clarity.

Abstract:
• Line 26-27: preposition
• Line 37-38: Punctuation
• Line 41: did you have more than one OP group?

Suggestions for the manuscript:
1) In lines 53-55, the activation approaches of PRP are described. This section would benefit by including more details about the importance of platelet activation to ensure continuity and clarity.

2) In line 63, the advantages of using PRP in the clinical practice including safety and invasiveness should be expanded. onThe importance of PRP in treating thin endometrium was not clearly described. More details about its clinical use in reproductive medicine should be included and this information should be better connected to the research questions. More details are needed such as how PRP promotes tissue regeneration.

3) In lines 76-77, it would be better to justify the reason for using a rat model instead of a human model (accessibility is a good argument). Also, MSC is usually isolated from bone marrow & adipose tissue; justify the reason for using uterine derived MSCs.

Experimental design

The method section is well defined and detailed with enough information to be reproducible. The statistical tests used are suitable for the generated data. However, looking at the raw data file, there is considerable variation of western blot data in the technical triplicate. This should be explained. Was the experiment repeated or optimized?

Method section:
1. The ethical approval sentence is repeated (lines 86-87 & 93-94).
2. In line 102, how much blood was collected?
3. Western blot assay section: I suggest reporting the final antibody concentration or dilution especially if optimization was required during the study.

Validity of the findings

The results are reasonable giving the experiments.

Suggestions:

Results section

1.In line 230, the expression of p53 in OP treated cells is mentioned; what about the p53 protein level in RPP treated cells? This finding should be reported to reflect the figures.
2.In line 260: indicate the name of cultures (SF, SM).

Discussion section
Overall, the discussion section reads well but it could be developed further. Here are a few suggestions:
In line 288, discuss the autophagy and self-renewal in more detail. The argument is excellent but it needs to link the finding with what has been reported in the literature. Also, it should be supported by citations.

In line 311, discuss the meaning of the result. It has been shown that a high MMP-9 expression is implicated in vasodilation, placentation and uterine expansion during normal pregnancy and changes in the activity or expression may indicate pregnancy complications. How can PRP enhance the implantation and endometrial growth in women with a thin endometrium?

Conclusion

Try to reformulate this section by explaining what this study adds to the existing literature.

Reviewer 2 ·

Basic reporting

The scientific reporting of this work is sound. However, a few words need to be replaced (e.g. etc.) and are highlighted in the attached pdf file. In addition, the author(s) occasionally used a single study (that is not with high level of evidence) to make a general assumption in areas where there are compelling evidence.

Experimental design

no comment

Validity of the findings

The findings of the work support the current applications of PRP in clinical practice. However, the used statistical approach may need more sample size to empower the outcomes. Also, the method of determining the data distribution (normal or non-normal) should be disclosed.

Additional comments

This is a great piece of work and could be published once the necessary corrections are considered or addressed.

Annotated reviews are not available for download in order to protect the identity of reviewers who chose to remain anonymous.

---

## Round 0.2 · Minor Revisions

The authors properly addressed most of the reviewers' comments. Please address the remaining comments.

Reviewer 1 ·

Basic reporting

The article is well written.
The only weakness is the length of the introduction.Try to shorten it by omitting some details or reformat the paragraph (50-56 , 62-71, 106-111).

Experimental design

no comment

Validity of the findings

The results are reasonable giving the experiments.

Additional comments

The manuscript provied important data in the filed of regenerative medicine with an important addition to the literature.

Reviewer 2 ·

Basic reporting

The discussion and conclusion part are well written. The introduction is very extensive and better than the previous submission. However, the introduction requires a few adjustments (re-structure) to flow better. I put a few suggestions for you to do that. Also, It is utterly important to include the motivation and purpose of this study in the introduction.

Experimental design

No comment.

Validity of the findings

No comment.

Additional comments

Thanks for re-submitting the manuscript and doing the suggested modifications.
Please, work on re-organizing the introduction and include the aim and motivation of your research.

Annotated reviews are not available for download in order to protect the identity of reviewers who chose to remain anonymous.

·

Basic reporting

There are grammatical errors and instances of bad sentence construction throughout the manuscript which need to be reviewed and corrected. This should be improved to ensure that the manuscipt is easily readable and clearly understood by the audience.

Experimental design

No comments

Validity of the findings

The findings of this study corresponds with previous research findings. Can you expand as to the reasons why the percentage of positive cells for CD105 was lower than expected (line 314-315)

Additional comments

Overall, I think this is a good article,

---

## Round 0.3 · accepted · Accept

Thank you for your contribution!

Reviewer 1 ·

Basic reporting

Overall, the manuscript is now well written. The use of English language is better and the introduction section follow clearly.

Experimental design

no comment

Validity of the findings

no comment

·

Basic reporting

No Comment

Experimental design

No comment

Validity of the findings

No comment

Additional comments

This article is well written and the revisions are satisfactory